# Machine learning identifies candidates for drug repurposing in Alzheimer's disease

Steve Rodriguez[1,2,3], Clemens Hug [1,3], Petar Todorov [1], Nienke Moret [1], Sarah A. Boswell [1], Kyle Evans[1,2], George Zhou [1,2], Nathan T. Johnson [1], Bradley T. Hyman [2], Peter K. Sorger [1], Mark W. Albers [1,2✉] & Artem Sokolov [1✉]

Clinical trials of novel therapeutics for Alzheimer's Disease (AD) have consumed a large amount of time and resources with largely negative results. Repurposing drugs already approved by the Food and Drug Administration (FDA) for another indication is a more rapid and less expensive option. We present DRIAD (Drug Repurposing In AD), a machine learning framework that quantifies potential associations between the pathology of AD severity (the Braak stage) and molecular mechanisms as encoded in lists of gene names. DRIAD is applied to lists of genes arising from perturbations in differentiated human neural cell cultures by 80 FDA-approved and clinically tested drugs, producing a ranked list of possible repurposing candidates. Top-scoring drugs are inspected for common trends among their targets. We propose that the DRIAD method can be used to nominate drugs that, after additional validation and identification of relevant pharmacodynamic biomarker(s), could be readily evaluated in a clinical trial.

[1] Laboratory of Systems Pharmacology, Harvard Program in Therapeutic Science, Harvard Medical School, Boston, MA, USA. [2] Department of Neurology, Massachusetts General Hospital, Charlestown, MA, USA. [3] These authors contributed equally: Steve Rodriguez, Clemens Hug. ✉email: albers.mark@mgh.harvard.edu; artem_sokolov@hms.harvard.edu

Alzheimer's disease (AD) is a growing healthcare crisis with longer life expectancy as its principal risk factor. It is estimated that, in the absence of effective prevention and treatment options, disease prevalence will more than double over the next several decades: from 5.8 million individuals living with AD today in the US to a projected 13.8 million by 2050 (ref. [1]). In addition to its direct impact on human health and welfare, the long-term care of affected individuals imposes a substantial economic burden[2]. Multiple efforts to develop disease-modifying therapeutics for AD, including approximately 200 clinical trials to date, have been largely negative with many failures occurring due to lack of efficacy or excess toxicity[3]. Every failed clinical trial of a new molecular entity (NME) consumes substantial time and resources. In contrast, repurposing drugs already approved by the Food and Drug Administration (FDA) for a different indication is less expensive, involves already defined possible toxicities, and can have a higher success rate (30%) as compared to development of a NME[4].

The traditional approach to repurposing is to use an existing drug in a new indication, perhaps at a different dose or in different formulation[5]. However, an alternative is to use repurposing as a way of testing a therapeutic concept that could then subsequently be advanced, with additional medicinal chemistry and functional testing, to become a NME. This is potentially valuable in the case of AD in which the underlying disease mechanisms remain poorly understood and the potential for multiple distinct disease drivers exists. Repurposing drugs for AD has received increasing attention[6,7], but approaches to date have been largely hypothesis-driven, based on overlap between an existing pharmacological mechanism of action (MOA) and a putative disease-causing mechanism[8] or results of a clinical trial[9,10]. While some of these leads are promising, no successes have been reported to date.

For simplicity, we use the term drug in this work to broadly refer to both FDA-approved chemical entities as well as clinically tested compounds and drug-like pre-clinical molecules (often referred to as chemical probes). As databases of drug information grow, informatics-based approaches to drug repurposing have emerged. Tools such as MeSHDD[11], PREDICT[12], Rephetio[13], Connectivity Map[14], and others[15–17] seek to establish large-scale associations between perturbations induced by drugs and by disease processes, which can then be mined for repurposing opportunities. One drawback of existing repurposing tools is that they are rarely disease specific and include data on drug mechanisms and disease pathways (typically transcriptional or proteomic signatures) obtained from diverse biological settings, often with a strong bias toward cancer cell lines and biopsies. This is potentially problematic in the case of a disease such as AD that is poorly understood, characterized by phenotypic[18] and pathological heterogeneity[19], and involves non-proliferating cells.

We therefore sought to develop a repurposing approach that made combined use of -omic datasets on drug-induced perturbation of neuronal cells and molecular changes that occur in the brains of individuals suffering from different stages of AD, as collected by the Accelerating Medicines Partnership - Alzheimer's Disease (AMP-AD) effort[20]. We focused on gene expression measurements, because they provide a natural connection between preclinical cell culture platforms, where perturbation experiments can be carried out, and patient-derived tissue specimens. Gene expression also features prominently in previous drug repositioning efforts focused on aging[21] and AD[16,22], with several approaches making use of the Connectivity Map[14] and related tools[23,24] to identify compounds that induce similar transcriptional changes in a given disease context[25].

The approach we developed, drug repurposing in Alzheimer's disease (DRIAD), involves a machine learning framework that quantifies the association between the stage of AD (early, mid, or late) as defined by Braak staging[26] and any biological process or response that can be characterized by a list of gene names. Data characterizing AD were obtained from AMP-AD datasets and comprised mRNA expression profiles of postmortem brain specimens. In parallel, lists of gene names were obtained by using RNAseq to measure the responses of human neuronal cells to small molecule drugs and then identifying differentially expressed genes to generate drug-associated gene lists (DGLs). In the current work, we focus on kinase inhibitors because they are associated with strong transcriptional signatures and have relatively well annotated target spectra[27]. DRIAD uses a specific DGL for feature selection and then trains and evaluates a predictor of AD pathological stage from AMP-AD gene expression data. In this way, the relevance of a drug-induced perturbation of neurons and other neural lineage cells grown in culture (which is a reflection of drug mechanism(s) of action) to the pathological processes underlying AD can be evaluated. Drugs whose DGLs resulted in the most accurate predictors were found to target proteins in signaling networks regulating innate immunity, autophagy, and microtubule dynamics; these represent previously unexplored pathways for a potential Alzheimer's therapeutic[28]. The direction of this effect is not specified a priori and DRIAD will identify both disease-enhancing and disease-reducing drugs, a topic addressed in the discussion. We show that DRIAD is agnostic to the length of the input gene list and its source, which can be a drug-induced perturbation, as in this work, the results of a previous study, or manual annotation of biological mechanisms.

## Results

**Machine learning framework for identifying potential associations between gene lists and disease.** When used for AD repurposing, DRIAD requires two types of inputs: (i) mRNA expression profiles from human brains at various stages of AD progression and (ii) a dataset comprising DGLs—lists of genes differentially expressed when neuronal cells (in the current work, a mixed population of neurons, astrocytes, and oligodendrocytes, differentiated from an immortal neuroprogenitor line) are exposed to a test panel of drugs. Human brain gene expression levels were taken from AMP-AD datasets[20] provided by The Religious Orders Study and Memory and Aging Project (ROSMAP), The Mayo Clinic Brain Bank (MAYO), and The Mount Sinai/JJ Peters VA Medical Center Brain Bank (MSBB), each encompassing measurements from one or more regions of the brain (Fig. 1b). Braak staging scores[26], assigned through neuropathological assessment of neurofibrillary tangle accumulation, were used to group samples into three categories of disease progression: early (A; stages 1–2), intermediate (B; stages 3–4), and late (C; stages 5–6). This grouping recapitulates the spatio-temporal progression of neurofibrillary tangles from the entorhinal region to the hippocampus area and, subsequently, to neocortical association areas[29].

DRIAD trains and then evaluates a predictor that can recognize the A, B, or C disease categories from mRNA expression levels, restricting the predictor to those genes in the DGL (Fig. 1a) (see "Methods"). It is likely that the molecular mechanisms involved in initiation and progression of AD are obscured in end-stage RNAseq profiles as a result of the actions of myriad signaling pathways and feedback loops, leading to widespread transcriptional changes only indirectly associated with disease mechanism[30–33]. This is reflected in the fact that many randomly selected list of genes in human brain gene expression profiles is weakly predictive of disease stage (Supplementary Fig. 1). We therefore sought lists of genes that

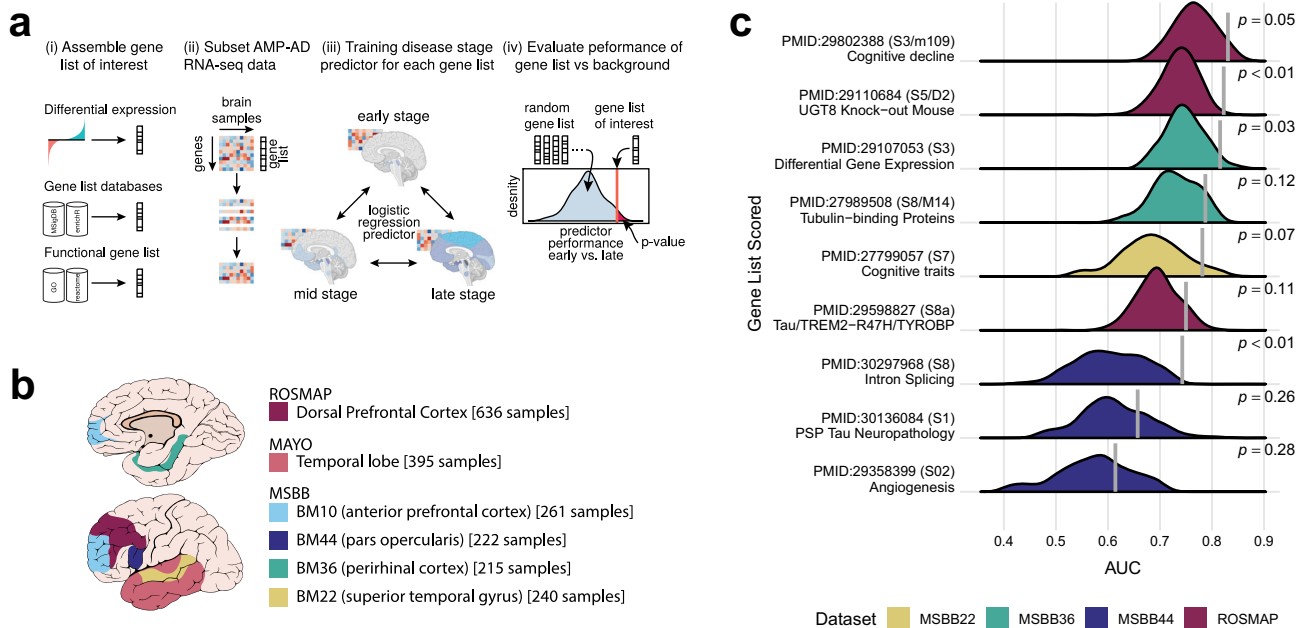

**Fig. 1 The definition and validation of the DRIAD framework. a** Overview of the machine learning framework used to establish potential associations between gene lists and Alzheimer's disease. (i) The framework accepts as input gene lists derived from experimental data or extracted from database resources or literature. (ii) Given a gene expression matrix, the framework subsamples it to a particular gene list of interest, and (iii) subsequently trains and evaluates through cross-validation a predictor of Braak stage of disease. (iv) The process is repeated for randomly selected gene lists of equal lengths to determine whether predictor performance associated with the gene list of interest is significantly higher than what is expected by chance. **b** AMP-AD datasets used by the machine learning framework. The three datasets used to evaluate the predictive power of gene lists are provided by The Religious Orders Study and Memory and Aging Project (ROSMAP), The Mayo Clinic Brain Bank (MAYO), and The Mount Sinai/JJ Peters VA Medical Center Brain Bank (MSBB). The schematic highlights regions of the brain that are represented in each dataset. The MSBB dataset spans four distinct regions, which are designated using Brodmann (BM) area codes. **c** Performance of predictors trained on gene lists reported in previous studies of AMP-AD datasets. The predictors are evaluated for their ability to distinguish early-vs-late disease stages with performance reported as area under the ROC curve (AUC). The vertical line on each row denotes predictor performance associated with a gene list reported in the literature, while the background distribution is constructed over randomly selected lists of matching lengths. Each row is annotated with the pubmed ID of the study, the supplemental resource that contained the gene list, and a short keyphrase providing functional context. Shown unadjusted p-values were computed with a one-sided empirical test, by counting the fraction of randomly selected lists in the background distribution that outperformed the corresponding literature lists.

outperformed random lists in predicting AD stage at a pre-specified level of significance. Statistical significance was assessed by repeatedly sampling the space of all gene names to create a background distribution of random gene lists (of the same length) against which to evaluate a DGL (Fig. 1a). An empirical p-value was provided by the fraction of random gene lists that outperformed the DGL in the prediction task.

**Validation of DRIAD using gene lists associated with AD pathophysiology**. To test the DRIAD framework, lists of gene names previously reported in the literature to be associated with an aspect of AD progression[32,34–41] were substitute for DGLs (Supplementary Data 2). Each input gene list was used to train DRIAD to distinguish between early (A) and late (C) stage disease based on mRNA levels in AMP-AD data. We found that most of the published lists of AD-associated genes outperformed randomly selected lists of equal length (Fig. 1c). Thus, DRIAD effectively recapitulates previous attempts to identify genes and co-expression modules associated with disease severity. Similar results were obtained with human gene sets and mouse homologs previously reported to be associated with AD. For example, McKenzie et al. used AMP-AD data to establish that UGT8 is a key regulator of oligodendrocyte function and reported a list of genes that are differentially expressed in the frontal cortex of a UGT8 knockout mouse model[40]. DRIAD shows that these differentially expressed genes have a significant association with

disease severity in AMP-AD data (Fig. 1c), providing further evidence that UGT8 may play a role in neurodegeneration in humans.

Similarly, Mostafavi et al. identified a co-expression module of 390 genes that has a strong association with cognitive decline[34]. DRIAD confirms that a predictor trained to recognize pathological stage based on these 390 genes performs significantly (empirical p-value = 0.05) better than equivalent predictors trained on arbitrary sets of 390 genes chosen at random (Fig. 1c). In contrast, when we examined gene lists associated with Progressive Supranuclear Palsy (PSP) Tau Neuropathology and Angiogenesis, DRIAD found that they correlated with Braak staging no better than random gene lists. This suggests that models trained using DRIAD can distinguish molecular features of tauopathies and pathologic processes not known to be relevant to AD. Consistency in performance was observed across datasets and brain regions, with one exception: whenever a predictor was trained on late-stage samples in the MAYO dataset, it substantially outperformed similar predictors evaluated on all other datasets (Supplementary Fig. 1). This suggests the presence of a strong batch effect in the late-stage MAYO samples that the predictors pick up in lieu of learning disease severity. Because the nature of this batch effect is unknown, we chose to exclude MAYO data in the current study. However, future studies could adjust for batch effects of unknown origin using established methods from the literature[42–44].

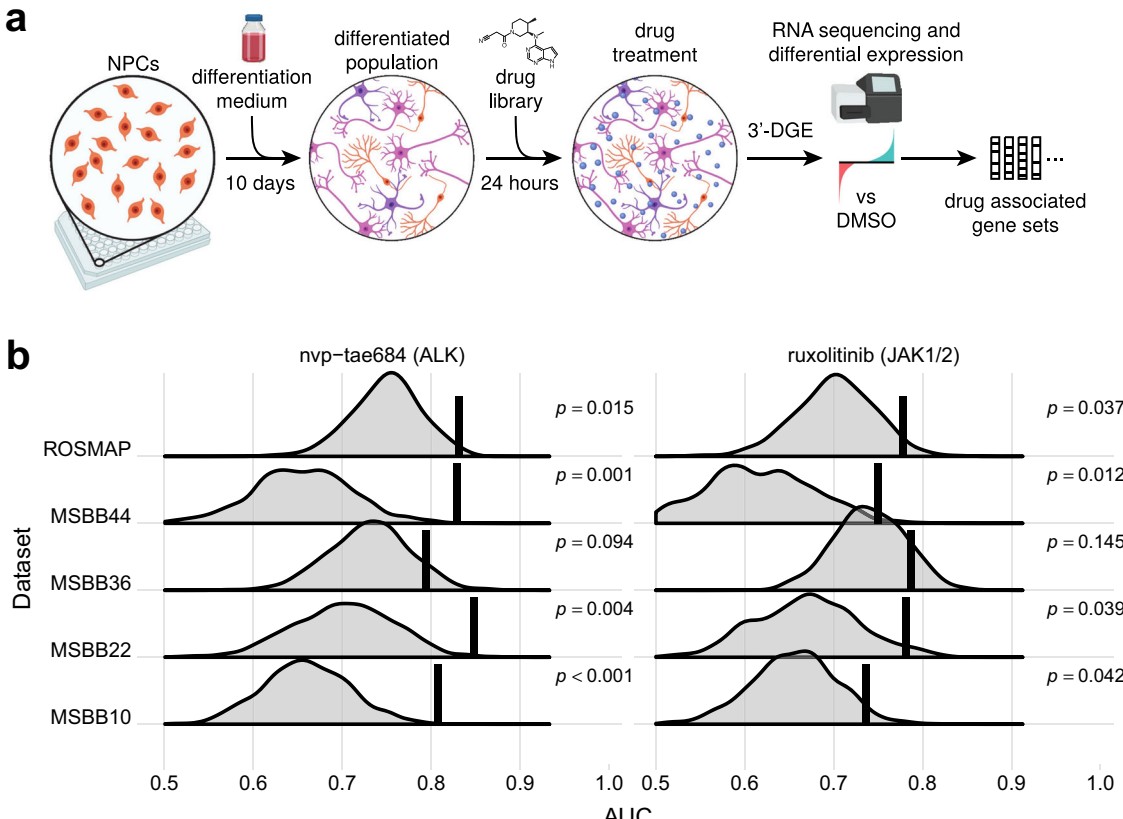

**Fig. 2 Collection and evaluation of drug-associated gene lists. a** Overview of the 3′ DGE experimental protocol used to derive drug-associated gene expression signatures. ReNcell VM human neural progenitor cells were plated and differentiated for 10 days, resulting in a mixed cell population of neurons, glia, and oligodendrocytes. The mixed culture was subsequently treated with a panel of drugs (Supplementary Data 3) at 10 μM for 24 h and frozen in a lysis buffer until library preparation. RNA was extracted and reverse transcribed into cDNA in each well of the plate, followed by pooling and preparation of mRNA libraries. After sequencing, mRNA reads were demultiplexed according to well barcodes, and the resulting gene expression profiles were processed by a standard differential expression method to derive drug-associated gene lists. **b** A highlight of two compounds whose gene lists consistently yield improved performance over the randomly selected lists of equal length. Shown is performance associated with predicting early-vs-late disease stages in several AMP-AD datasets. Each row corresponds to an evaluation of gene lists in a single dataset; MSBB evaluation is subdivided into four brain regions, specified as Brodmann Area. The vertical line denotes performance of the drug-associated list, while the background distribution shows performance of gene lists randomly selected from the same dataset. The drugs are annotated with their nominal targets. The unadjusted *p*-values were computed with a one-sided empirical test, by counting the fraction of randomly selected lists that outperformed the corresponding drug-associated lists.

**3′ Digital Gene Expression profiles drug-induced perturbation of mRNA expression**. Drug responses were generated using the RenCell VM human neural progenitor cell line. Upon growth factor withdrawal, RenCell VM cells differentiate into a mixed culture of neurons, glia, and oligodendrocytes over a period of ~10 days[45]. Differentiated RenCells were exposed for 24 h to one of 80 kinase inhibitors at two doses in triplicate, or to a vehicle-only (DMSO) control, and mRNA levels were then measured using the high-throughput, intermediate read density RNAseq method 3′ Digital Gene Expression (3′ DGE)[46] (Fig. 2a). The advantage of 3′ DGE in this setting is that it provides high-quality gene expression signatures at relatively low cost, allowing data to be collected from multiple repeats, doses, and drugs. The small molecule panel was designed to include FDA-approved drugs, which could potentially be repurposed, compounds that have been tested in human clinical trials—and for which toxicity data are available—that could be further developed for use in AD, as well as pre-clinical compounds designed to extend the range of targets and test therapeutic concepts not explorable with clinical grade compounds (Supplementary Data 3). Kinase inhibitors were chosen as a class of well-annotated drugs that target many different members of a single gene family and elicit strong transcriptional signatures. The 80 kinase inhibitors were profiled

across two separate 3′ DGE experiments and, to establish reproducibility, 5 of the 80 compounds were included in both experiments. The experiments are indexed separately (Supplementary Data 3), and the compounds in common provide a measure of biological and technical variation. High concordance of the measurements between the two experiments was observed (Supplementary Fig. 2) suggesting that batch effects were not strong. Overall, the drug-response data comprised 767 DGE gene expression profiles.

For each drug, we defined the DGL to be the set of significantly perturbed genes, as identified through differential gene expression analysis[47] comparing 3′ DGE profiles of drug-treated and control cells (see "Methods"). For most drugs, we identified between 10 and 300 significantly perturbed genes. DRIAD evaluated each DGL by constructing a predictor of pathological stage based on mRNA expression of these genes in AMP-AD datasets. The accuracy of the predictor assessed over multiple brain regions measures the association between a drug's mechanism (encapsulated in the DGL) and AD severity. We focused specifically on the binary classification task of distinguishing early vs. late disease stages (A-vs-C), because it contrasts groups of maximally distinct samples and yields higher signal to noise ratio than attempting to predict all three disease stages. This caused random gene lists to

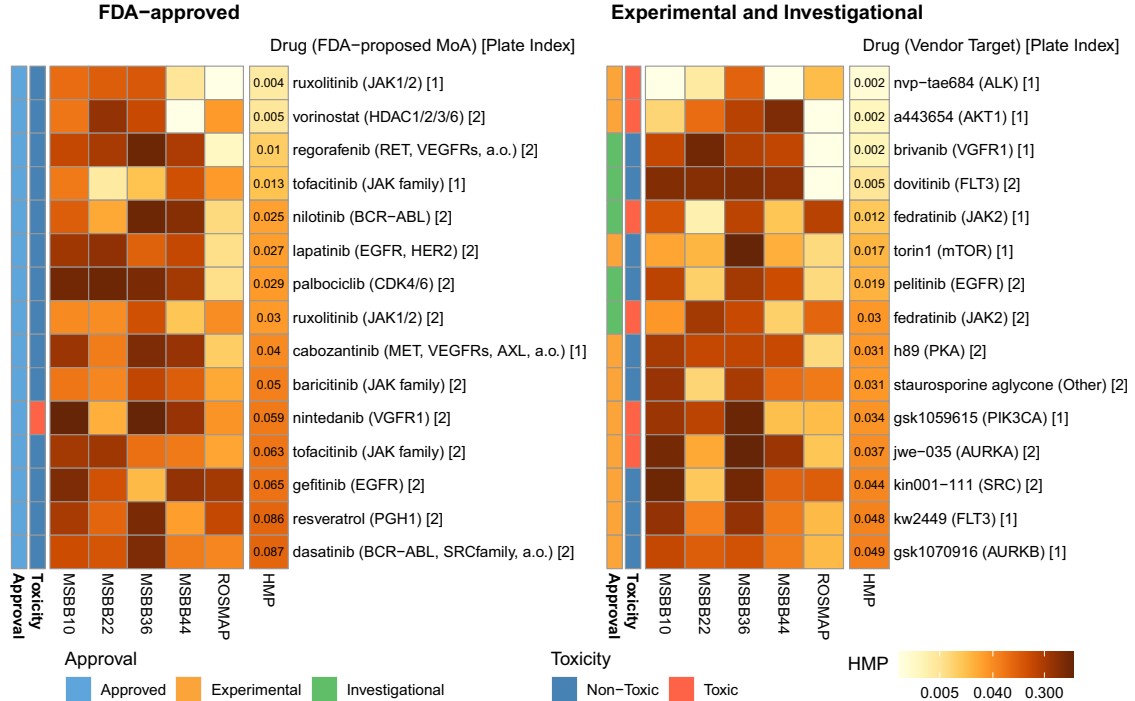

**Fig. 3 Top 15 FDA-approved (left) and experimental/investigational (right) drugs, sorted by harmonic mean *p*-value.** Each heatmap shows unadjusted empirical *p*-values associated with a drug's predictive performance across two AMP-AD datasets, ROSMAP and MSBB. The MSBB analysis is further subdivided by the brain region, specified as Brodmann Area. The empirical *p*-values were computed by counting the proportion of randomly selected lists that outperformed the gene lists of interest (i.e., a one-sided test). The *p*-values were then aggregated across the datasets by computing the harmonic mean *p*-value (HMP), which is shown in the last column of each heatmap. The rows are annotated with the name of the drug/compound, its nominal target, and the index of the corresponding DGE experiment. Additional annotations include information about each compound's approval status (approved/investigational/experimental) and whether compounds were found to be toxic in neuronal cell cultures.

have higher predictive power, creating a higher bar for DGLs to surpass to be considered significant.

**Systematic assessment of drug signatures derived from 3′ DGE leads to a ranked list of repurposing candidates.** Each DGL is evaluated against randomly selected gene lists of the same length (Fig. 1a). The empirical *p*-value, computed as the fraction of random lists that outperform a DGL, constitutes a natural starting point for comparing predictor performance across drugs, because it is effectively normalized by the number of genes in the DGL. An example of drugs whose DGL consistently outperform random lists include the pre-clinical compound TAE684 and the approved drug ruxolitinib, whose primary targets are ALK and JAK1/2 kinases, respectively (Fig. 2b). Next, we combined the empirical *p*-values from multiple datasets and brain regions to create a harmonic mean *p*-value (HMP) for each drug. The HMP facilitates the detection of significant hypothesis groups in a multiple hypothesis setting, while being less restrictive and providing greater statistical power than similar multiple hypothesis testing procedures[48]. Using HMP as a prioritization scheme, we identified the top 15 FDA-approved and top 15 pre-clinical drugs (Fig. 3) in the full ranking of all 80 drugs that were profiled in differentiated RenCell cultures (Supplementary Data 3).

Some of top-performing drugs were developed as antineoplastics and are known to be cytotoxic in some cell types (e.g. TAE684 (ref. [49])). To assess the magnitude of this cytotoxicity, we used fixed cell microscopy to quantify the fraction of differentiated RenCells that were viable following 48 h of exposure to a high drug dose (10 μM; see "Methods" and Supplementary Fig. 3). Drugs that significantly reduced viable cell number were annotated cytotoxic (Fig. 3). We also observed

that viable cell number was correlated with total RNA yield (Wilcoxon Rank Sum test, $p < 4 \times 10^{-9}$), suggesting that compound toxicity could also be inferred from a reduction in mRNA abundance in post-perturbational gene expression profiles (Supplementary Fig. 3). A cytotoxic drug does not seem a promising lead for an AD therapeutic but since DGLs of cytotoxic drugs consistently outperform random lists across multiple AMP-AD dataset (Figs. 2b and 3), the mechanisms of cell death induced by these drugs may share some similarity with mechanisms of cell death in AD, potentially leading to hypotheses that could be tested in follow on studies targeting cell death mechanisms in neurodegeneration.

**Elucidating target affinity spectrum properties associated with the observed drug ranking.** Do high-scoring drugs have features in common? With respect to primary targets, we observed that many drugs, including ruxolitinib, inhibit one or more of members of Janus Kinase family, which comprises JAK1, JAK2, JAK3, and TYK2 (Fig. 3). However, these compounds are known to have additional off-targets that might also contribute to activity[27,50]. To investigate the potential role of these secondary targets we used the target affinity spectrum (TAS), a vector of activities against a range of targets computed from experimental data that quantifies the potency of a drug based on a variety of biochemical assays[27]. TAS vectors were constructed by aggregating information about targets and non-targets from published dose–response measurements, experimental profiling data involving multiple targets assayed in parallel, and manual annotations in the literature. Confirmed drug–target interaction was assigned a TAS score of 1, 2, or 3 (with lower values indicating higher binding affinity), while confirmed non-binders were annotated with a

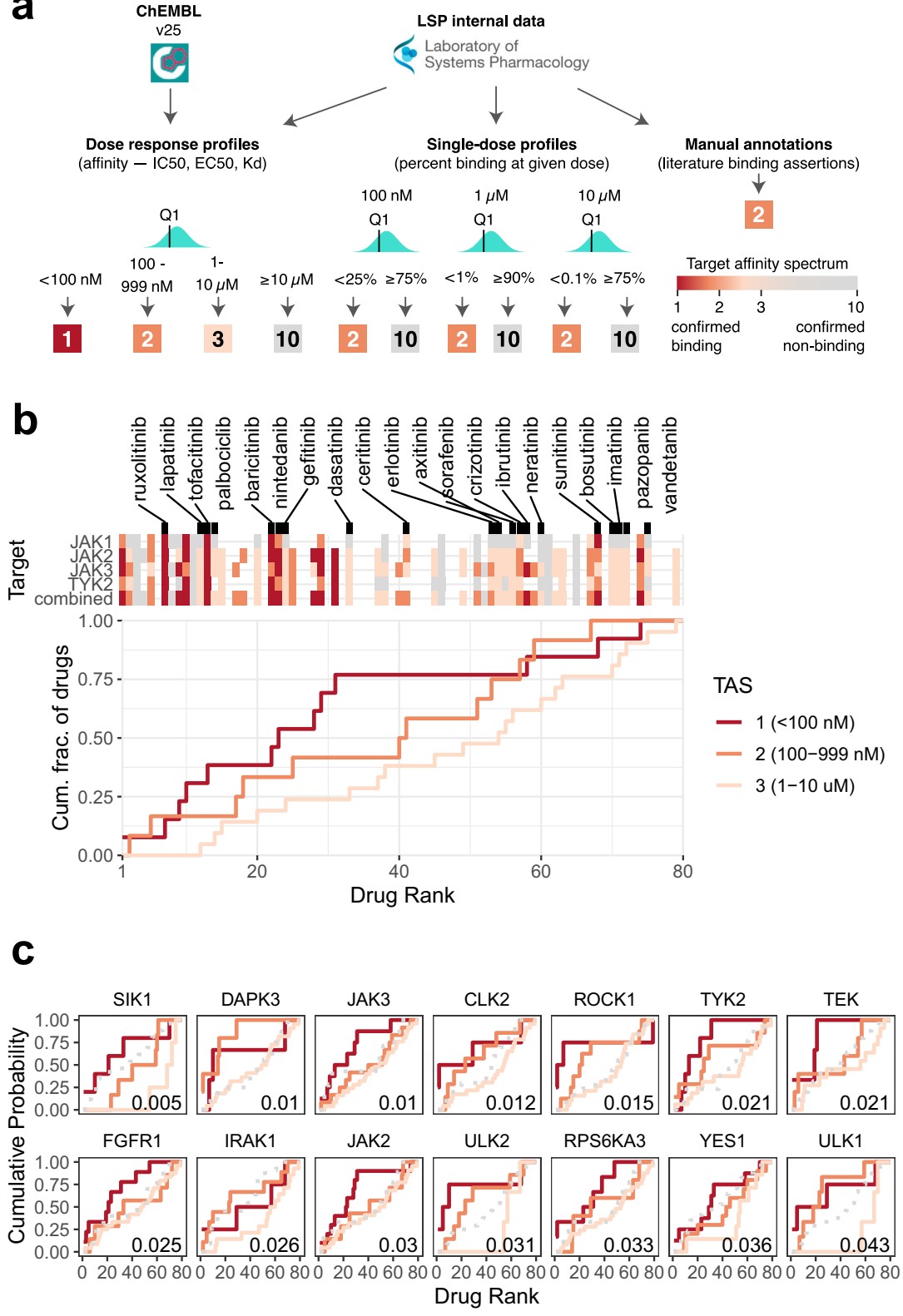

TAS score of 10 (Fig. 4a). While less precise than affinity measurements, TAS assertions offer a more complete picture of known targets and known non-targets, enabling more comprehensive target discovery for high-scoring drugs. This is important in the current work because virtually all kinase inhibitors, including clinical grade drugs, have multiple targets, and the best

known target, or the one listed in an FDA approval, is not necessarily the highest affinity binder. The full set of TAS scores, which includes the 80 compounds considered in this study, is publicly available at http://smallmoleculesuite.org.

We evaluated whether the strength of a binding interaction (i.e., binding affinity) between a compound and its target or a set

**Fig. 4 Analysis of target affinity among the top-scoring drugs. a** Overview of target affinity spectrum (TAS) score computation from raw drug binding data. Three types of drug binding data were sourced from ChEMBL and from the internal Laboratory Systems of Pharmacology dataset that have not yet been incorporated into ChEMBL. Empirically derived thresholds for the different data types were used to assign TAS scores to each drug–target pair. Multiple measurements for the same drug–target combination were aggregated along the first quartile to define the final TAS value. **b** Binding affinity of compounds in the ranked list to every member of the Janus Kinase family. The compounds are sorted in increasing order by the harmonic mean *p*-value (as defined in Fig. 3) along the *x*-axis. The top heatmap shows the binding affinity of each compound to the selected targets, explicitly naming the FDA-approved drugs. Colored and gray tiles denote confirmed binders and non-binders, respectively; missing entries correspond to unknown affinity values. The combined affinity is defined as the strongest binding (lowest TAS score) among all four JAK targets. The bottom plot shows the breakdown of the combined affinity values by TAS-specific empirical cumulative distribution functions (ECDFs). Each line shows ECDFs for all drugs that bind the corresponding target with a TAS score of 1 (dark orange), 2 (orange), or 3 (light orange). **c** Top targets whose binding affinity correlates most strongly with the compound ranking. The ECDFs of confirmed non-binders (TAS = 10) are shown as gray dashed lines for reference. Area under ECDF can be interpreted as a summary statistic that captures the position of drugs binding to that target with the corresponding affinity in the ranked list. Correlation between the drug ranking and TAS values was computed using the one-sided Kendall's Tau test, with the associated unadjusted *p*-value displayed in the bottom right corner of each plot.

of biologically related targets contributes to the ranking of a compound by DRIAD. Significant positive correlations between the drug ranking and the binding affinity suggest that the target or class of targets is likely to be pertinent to one or more disease mechanisms. To illustrate this effect, we constructed empirical cumulative distribution functions (ECDFs) for the binding of drugs to members of the JAK family. The list of 80 drugs was traversed in the order of increasing harmonic *p*-value, while keeping track of the cumulative count of drugs with a particular TAS value (Fig. 4b). This results in three different ECDFs, representing all drugs that bind to the corresponding target with a TAS score of 1, 2, or 3. Area under individual ECDF curves (AUC) can be interpreted as a summary statistic capturing the position of JAK binders with a particular affinity in the ranking; larger values of the AUC correspond to a higher saturation of the corresponding drug set near the top of the ranking.

We observed that compounds having higher affinity for members of the JAK family (i.e., lower TAS values) tend to appear earlier in the ranking (Fig. 4b), with a significant correlation (*p* = 0.001; Kendall's Tau test) between binding affinity and the drug ranking as defined by DRIAD. This suggests that downstream transcriptional changes induced by JAK inhibitors anticorrelate with Braak AD stage severity in an affinity-dependent manner. Direct inspection of the experimental 3′ DGE data confirms that binding affinity directly correlates with the levels of expression of several interferon-stimulated genes: drugs that have a higher affinity to JAK family members resulting in stronger inhibition of interferon gene expression (Supplementary Fig. 4). These observations are in accord with previous studies[51] suggesting that inhibition of the JAK-STAT and interferon signaling pathways might be beneficial in the context of AD.

We repeated the binding affinity correlation analysis for all targets that had confirmed binding interactions with at least three drugs used in this study (Fig. 4c). The results showed that most of the affinity-dependent effects were contributed by "off-targets", i.e., established targets for a drug that are different from the target for which the drug is marketed (the nominal target). For example, we observed strong associations between drug ranking and binding affinity for Unc-51-Like Kinases 1 and 2 (ULK1, ULK2) and their downstream substrate Death-Associated Protein Kinase 3 (DAPK3), all of which are associated with autophagy[52–54]. Autophagy plays an important role in cellular homeostasis and its dysregulation is emerging as a contributing factor to neurodegenerative diseases, including AD[55,56]. Previous studies suggest that inhibition of autophagy may impair clearance of neurotoxic aggregates. Thus, effective AD therapies may need to maintain or increase autophagy as part of their MOA (e.g. by perturbing proteins that function upstream)[57,58]. The Salt Inducible Kinase 1

(SIK1) also appears to have a strong association with the position of compounds in the ranking. This association is driven primarily by TAE684, fedratinib, and GSK1070916, all of which are strong binders of SIK1 (TAS = 1) and appear near the top of ranking established by DRIAD. However, none of these drugs are FDA approved or have been studied in the context of SIK1 inhibition.

**Polypharmacology analysis reveals additional mechanisms that may correlate with AD severity.** We also considered downstream effects of concerted off target binding (polypharmacology) in which genes with a closer association to disease severity are altered more significantly by coordinated activity on two or more targets relative to drugs that selectively bind to only one of the off-targets. To evaluate the impact of polypharmacology on drug–disease associations, we divided the 80 compounds in our dataset into three categories: those with confirmed binding to Target A and Target B and those that bind either Target A or Target B alone (see "Methods"). The three categories were then compared to determine whether compounds binding both targets appear significantly closer to the top of the ranking (Fig. 5a) as defined by the HMP value computed by DRIAD (Supplementary Data 3). We systematically evaluated all pairs of targets that had a least six compounds with confirmed binding interactions associated with TAS values of 1, 2, or 3, and then identified the top pairs in the ranking (Fig. 5b). A pair was deemed to be a positive interaction if compounds binding both targets (A and B) appeared significantly closer to the top of the ranking than compounds binding only one of the targets (A or B, but not both); the pair was deemed to be a negative interaction if the opposite was true.

To determine whether one or more targets consistently participate in positive or negative interactions with other targets, we aggregated individual *p*-values from the evaluation of target pairs using Brown's method and Jaccard similarity as the metric of independence between individual tests (see "Methods"). The list of targets was subsequently sorted by the aggregated *p*-value (Table 1 and Supplementary Data 1). We found that several gene families emerged as candidate for top-scoring compounds in which polypharmacology was predicted to be essential. For example, the top-scoring compounds TAE684, dovitinib, ruxolitinib, and fedratinib are observed to bind RPS6KA1 (Fig. 5a and Table 1), a component of a microglial signature[59] with a potential role in AD as identified by previous epigenetic studies[60], and RPS6KA2, a gene involved in Neurotrophin signaling[61] with previous reports of association with Parkinson's Disease in GWAS studies[62]. Similarly, NIMA-related kinases NEK3, NEK6, and NEK9 consistently appear in positive interaction with other drug targets among top-scoring compounds (Table 1). All three genes have known relationships to microtubule function and Tau phosphorylation. NEK3 has been reported to

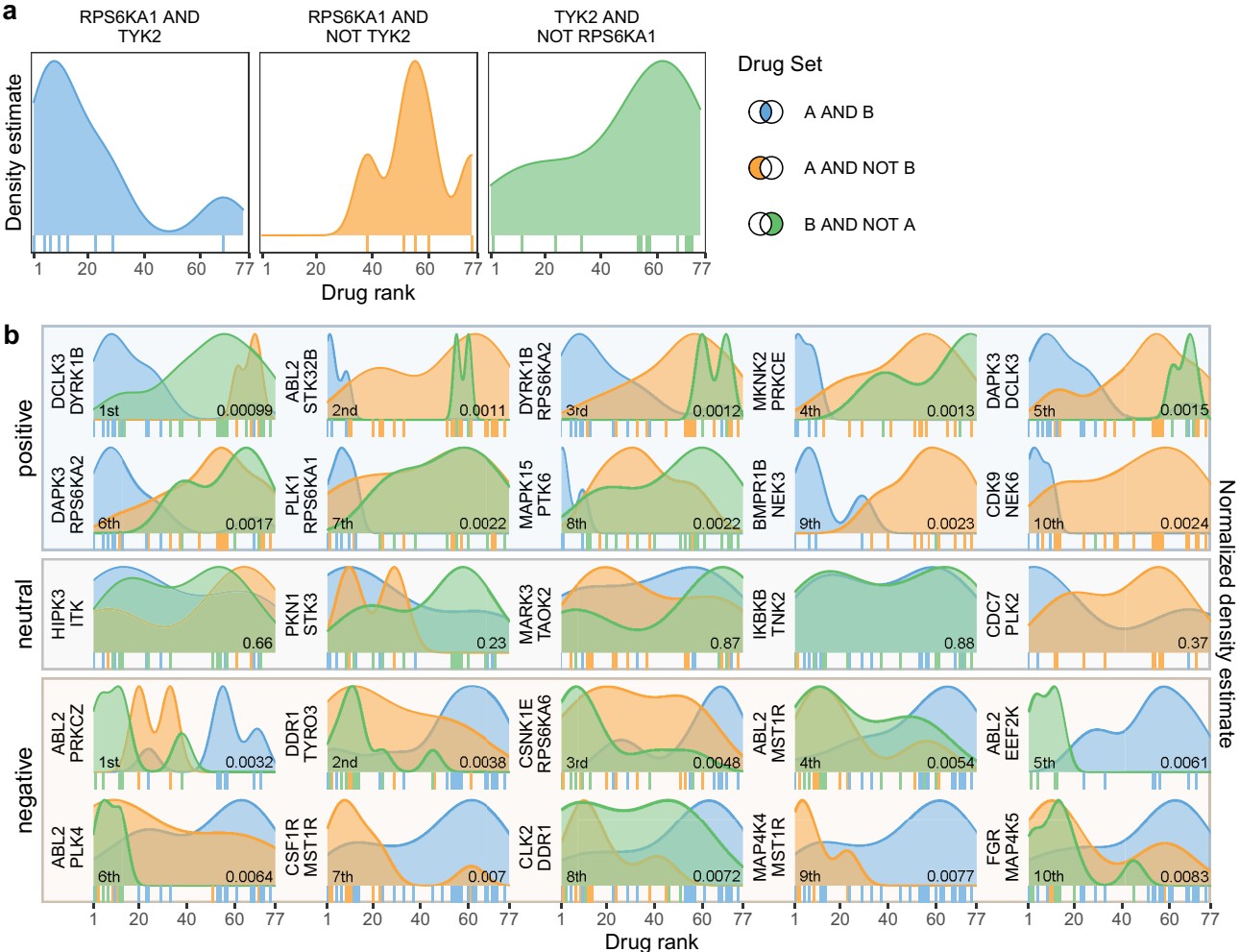

**Fig. 5 Analysis of polypharmacology effects among the top-scoring drugs. a** An example polypharmacology test with a focus on RPS6KA1 and TYK2. The drugs are ranked by the harmonic mean *p*-value (as in Figs. 3 and 4), and the distributions of drugs bindings to both RPS6KA1 and TYK2 (left), those binding to RPS6KA1 but not TYK2 (middle) and, conversely, TYK2 but not RPS6KA1 (right) are shown along this ranking. Individual drugs that bind those targets are annotated by vertical tick marks directly below the corresponding distribution. **b** Top ten positive and top ten negative interactions between pairs of targets. The distributions in each plot are compared using Wilcoxon Rank Sum test, with the resulting *p*-value presented in the bottom right corner. If compounds that bind both targets appear significantly closer to the top of the ranked list (left side of the *x* axis), we define the target pair to be a positive interaction. Conversely, a pair of targets with an explicit non-binding interaction observed among the top-ranking compounds is defined to be antagonistic. A set of five neutral target pairs (i.e., no significant positive or negative effect) is included for reference.

**Table 1 Top ten targets that consistently appear in the top positive or negative interactions.**

| Symbol | Direction | *n* | *p* | *p*$_{adj}$ |
| --- | --- | --- | --- | --- |
| NEK6 | Positive | 257 | 8.25E−129 | 1.32E−06 |
| NEK3 | Positive | 237 | 5.45E−71 | 2.52E−04 |
| RPS6KA2 | Positive | 246 | 7.62E−68 | 4.80E−04 |
| LATS2 | Positive | 280 | 3.82E−60 | 9.53E−04 |
| ABL2 | Negative | 204 | 2.88E−51 | 9.94E−04 |
| DCLK3 | Positive | 250 | 1.23E−56 | 1.24E−03 |
| MARK1 | Positive | 271 | 4.12E−52 | 1.92E−03 |
| STK17B | Positive | 264 | 1.22E−49 | 2.21E−03 |
| NEK9 | Positive | 295 | 5.14E−50 | 2.66E−03 |
| STK17A | Positive | 202 | 8.57E−37 | 3.23E−03 |

The table lists targets (symbol), whether those pairs are primarily positive or negative interactions (direction), the number of pairs they appear in (*n*), and the overall *p*-value computed by aggregating *p*-values from individual Wilcoxon Rank Sum tests (Fig. 5b) using the Brown's method. Additional adjustment for multiple hypothesis testing was performed using the Benjamini Hochberg method (*p*$_{adj}$).

influence neuronal morphogenesis through microtubule acetylation, and its phospho-defective mutant is hypothesized to play a role in axonal degeneration[63]. NEK6 phosphorylates p70-S6 (ref. [64]), a key player in hyperphosphorylation of Tau[65,66] that leads to microtubule disruption and deposition of Tau tangles. This was found to be relevant to the progression of AD and proposed as an early diagnostic biomarker[67,68]. NEK9 was found to be differentially expressed in a tauopathy mouse model[69]. Taken together, these results suggest that the NEK family may be an important set of co-targets, and a successful future therapeutic might require polypharmacology with respect to these kinases.

## Discussion

In this paper, we described the development of DRIAD, a machine learning framework for evaluating potential relationships between a disease and any biological process that can be described by a list of genes. We used DRIAD to look for associations between the pathological stage of AD and genes

that are differentially expressed when a small molecule drug is applied to a culture of terminally differentiated neuronal progenitor cells, which comprises a mix of neurons, astrocytes, and oligodendrocytes. DRIAD is distinct from the traditional approaches in which a model is constructed over the entire gene space and subsequently interrogated for feature importance scores and the enrichment of predefined gene sets, which then serve as a candidate list for further functional studies[70]. Traditional approaches are well-suited for predictors that exhibit high accuracy, because they establish a strong association between input features and the predicted phenotype. As predictor accuracy decreases, however, it becomes difficult to distinguish whether a high enrichment score is a true association between the corresponding mechanism and disease, or an artifact in a model that does not accurately predict the phenotype of interest (here, disease stage as defined by Braak score). DRIAD effectively decouples gene set enrichment and predictor performance by filtering the transcriptomic space for genes associated with drugs prior to model training and predictor evaluation. Pre-filtering to a limited set of features also addresses issues with overfitting that often arise when constructing computational models from disease databases where the number of cases is much less than the number of 'omic features. Thus, DRIAD enables a direct, unbiased quantification of the association between the effects of a drug and AD progression.

The 80 compounds that we profiled in vitro were predominantly kinase inhibitors with anti-cancer activity since kinase inhibitors are the largest class of targeted drugs currently available, both as approved and pre-clinical compounds[27], with extensive target information and diversity in chemical structure (Supplementary Fig. 5). An observation of possible significance is that there exists an inverse relationship between incidence of cancer and incidence of AD[71,72]. Among the 80 compounds tested, 33 were FDA-approved (Supplementary Data 3) and can potentially be used for repurposing directly. The remaining set consisted of 33 pre-clinical and 14 investigational compounds, which allowed us to explore a wider spectrum of mechanisms. Targets of pre-clinical and investigational compounds that were scored highly by DRIAD could potentially be used for selection of FDA-approved compounds in future screens or for the development of NMEs.

We ranked all compounds by how well their MOAs (as represented by a list of gene names) were able to predict disease severity based on gene expression in AMP-AD datasets. We found several drugs whose primary targets are JAK kinases to be among the top performers. We also explored connections between drug and their primary and secondary targets. This revealed that top-ranking drugs modulate pathways related to interferon signaling, autophagy, and microtubule formation and function. Kinases from JAK, ULK, and NEK families were found to be among the most consistent targets of top-scoring drugs. Future investigation will include experimental validation of these targets in cell-based and animal models using CRISPRa and/or CRISPRi, or other gene editing techniques, to evaluate whether a drug "hit" from DRIAD has an impact on AD-associated pathophysiology.

DRIAD has the potential to identify drugs that both mimic (or accelerate) disease and those that inhibit it. From the perspective of actual drug repurposing, only the later compounds are useful. In other studies, gene expression changes have demonstrated increased interferon signaling in AD[73] and in ALS[74] brains. Recently, we have found that cytoplasmic dsRNA, a known activator of Type I interferon, is present in ALS brains with TDP-43 pathology. Similarly, cytoplasmic dsRNA, which has been linked to increased Type I interferon signaling, was found to accumulate in glia in AD brains[75]. Activation of interferon signaling in this context promotes neuronal cell death. Thus, it seems probable that the inhibitors of JAK-STAT signaling identified in this study will potentially be useful in blocking neuroinflammation and cell death in the context of AD. Further studies to investigate the role of JAK-STAT signaling in aging and in AD brains are therefore warranted.

DRIAD allows for unbiased assessment of biological processes or drug candidates even when disease mechanisms are not explicitly known. This is valuable from a neuropathological perspective because it is increasingly clear that in addition to the classical AD hallmarks of amyloid plaques, neurofibrillary tangles, and neuronal loss, most patients with a clinical diagnosis of AD dementia have distinct patterns of co-occurring pathologies including TD-P43 inclusions, Lewy bodies, vascular changes, astrocyte and microglial activation, and probably other unrecognized alterations[30,76–79]. By working directly with the mRNA expression data from postmortem brain specimens and a priori knowledge of which genes encapsulate a proposed mechanism or co-expression module, or which genes are perturbed by a set of drugs, DRIAD can score mechanisms, co-expression modules or drugs without explicit knowledge of co-existing pathologies, such as the presence of Lewy Bodies or TDP-43 inclusions, in individual patients. Thus, DRIAD is capable of evaluating diverse hypotheses, including those associated with repurposing, without a high level of prior knowledge.

The AMP-AD Knowledge Portal is the most comprehensive database of gene expression profiles from AD brains, combining data from multiple large-scale studies. The gene expression profiles from autopsied brains are associated with Braak pathologic stage, making it possible to compare patients with no or mild AD symptoms at the time of death to patients who were demented. However, this anchoring on Braak staging also includes some cases in which symptoms did not correlate with pathological stage. A follow-on computational approach to deconvolve the correlative signals observed between top-performing drug signatures and AD expression profiles would help inform subsequent mechanistic studies. One direction for follow-up is to rerun the DRIAD pipeline on patient subgroups as defined by more detailed clinical or pathologic phenotypes, motivated by the notion of personalized treatment: different molecular pathways of AD arising in different patients will likely require different interventions to rescue neuronal death. Additional directions for follow-up include applying DRIAD to other studies that include age-matched individuals with no evidence of AD pathology as controls to build a predictor for the risk of AD pathology, and studies that include additional molecular modalities (e.g., mass spec proteomics, genome-wide association studies, etc.) and clinical co-variates (e.g., cognitive decline, comorbidities).

Our study has identified associations of gene perturbations by a subset of FDA-approved drugs and investigational compounds—largely kinase inhibitors—in human neural cells with gene perturbations in AD brain regions, but it has a number of limitations. We will extend this approach to small molecule drugs with a greater diversity of mechanisms of actions, e.g. GPRC inhibitors, as well as consider other types of drugs that target key AD genes. Our results require validation in relevant in vitro and in vivo AD model systems with amyloid plaques, neurofibrillary tangles, and neuronal death[80] to examine impacts on key pathologic features or through emulated clinical trials in electronic health records[81]. Another limitation is knowing if drugs cross the blood-brain barrier (BBB), which is important for their use in brain diseases. These compounds were not developed to treat diseases of the brain, and we find inconsistencies in the methodology and animal models used across studies to assess BBB penetration for each

compound. Theoretical approaches have been developed that aim to predict the penetration of drugs into the brain based on physical characteristics and chemical structure; however, the predictability is not generalizable across all compounds[82]. Therefore, the only way to know if a drug crosses the BBB is through empirical studies in patients with the disease of interest by quantifying free and total drug levels in cerebral spinal fluid by performing lumbar punctures in patients in a relevant age range who are already taking an FDA-approved drug of interest or as a pharmacokinetic outcome of a pilot clinical trial for compounds of interest with unknown BBB penetration.

## Methods

**High-throughput profiling using 3′ DGE**. A multiwell cell dispenser (catalog# 5840300, Thermo Scientific, Waltham, MA) with standard tubing (catalog# 24072670, Thermo Scientific, Waltham, MA) was used to plate 2500 neural stem cells (ReNcell VM, catalog# SCC008, Millipore, Billerica, MA) into each well of a 384-well cell culture plate (Perkin Elmer, Waltham, MA). Neural stem cells were differentiated into mature neural cells for 1 week and then treated with compounds (Supplementary Data 3) or DMSO using a D300 Digital Dispenser (Hewlett-Packard, Palo Alto, CA). D300 software was used to randomize dispensation of compounds. After 24 h, the cells were washed once with PBS using an EL405x plate washer (BioTek, Winooski, VT) leaving 5–10 µl of PBS behind in each well; 10 µl of 1× TCL lysis buffer (catalog# 1070498, Qiagen, Hilden, Germany) with 1% (v/v) β-mercaptoethanol was added per well, and the plates were stored at −80 °C until the RNA extraction was performed.

For RNA extraction, the cell lysate plate was thawed and centrifuged for 1 min at $1000 \times g$. Using a BRAVO (Agilent, Santa Clara, CA) liquid handler, the lysate was mixed thoroughly before transferring 10 µl to a 384-well PCR plate; 28 µl of home-made Serapure SPRI beads (GE Healthcare Life Sciences, Marlborough, MA) were added directly to the lysate, mixed and incubated for 5 min. The plate was transferred to a magnetic rack and incubated for 5 min prior to removing the liquid to aggregate the beads. The beads were washed with 80% ethanol twice, allowed to dry for 1 min, 20 µl of nuclease-free water was added per well, the plate was removed from the magnetic rack, and the beads were thoroughly resuspended. Following a 5-min incubation, the plate was returned to the magnetic rack and incubated an additional 5 min before transferring the supernatant to a fresh PCR plate. Five microliters of the RNA was transferred to a separate plate containing RT master mix and 3′ and 5′ adapters for reverse transcription and template switching (Soumillon, et al., 2014), and incubated for 90 min at 42 °C. The cDNA was pooled and purified with a QIAquick PCR purification kit according to the manufacturer's directions with the final elution in 21 µl of nuclease-free water. This was followed by exonuclease I treatment for 30 min at 37 °C that was stopped with a 20-min incubation at 80 °C. The cDNA was then amplified using the Advantage 2 PCR Enzyme System (Takara, Fremont, CA) for six cycles, and purified using AMPure XP magnetic beads (Beckman Coulter Genomics, Chaska, MN). Library preparation was performed using a Nextera XT DNA kit (Illumina, San Diego, CA) on five reactions per sample following the manufacturer's instructions, amplified 12 cycles, and purified with AMPure XP magnetic beads (Beckman Coulter Genomics, Chaska, MN). All primers used in library preparation are detailed in Supplementary Table 1. The sample was then quantified by qPCR and sequenced on a single Illumina NextSeq run with 75 bp paired end reads at the Harvard University Bauer Core Facility.

Raw RNA reads were aligned against a reference genome and quantified using the bcbio-nextgen single cell/DGE RNAseq analysis pipeline (https://bcbio-nextgen.readthedocs.io/). The pipeline consists of the following steps: (1) well barcodes and unique molecular identifiers (UMIs) were extracted from every RNA read; (2) all reads not within the edit distance of a single nucleotide from a predefined well barcode were discarded; (3) each extant read was quasialigned to the human transcriptome (version GRCh38) using RapMap[83]; (4) reads per well were counted according to UMIs[84], discarding reads with duplicate UMIs, weighting multi-mapped reads by the number of transcripts they aligned to and collapsing transcript counts to gene level by summing across all transcripts of a gene.

Differential gene expression analysis was performed with the R package edgeR 3.18.1. Compound-associated gene lists were composed from genes with a significant (FDR < 0.05) post-perturbation change in expression level compared to DMSO controls. For most compounds, this produced a list with 300 or fewer genes. The remaining few lists were capped at the top 300 genes to (a) make them more consistent with the vast majority of profiled drugs, allowing for fairer comparisons across compounds; (b) increase the sampling space for the corresponding background sets, and (c) help prevent overfitting that arises when the number of features is vastly larger than the number of samples (Supplementary Fig. 1).

**Prediction of disease stage**. Gene expression profiles of postmortem brain specimens along with the corresponding clinical annotations were downloaded from the Synapse portal at www.synapse.org/AMPAD. The entire transcriptional feature space was filtered down to ~20k protein-coding genes to ensure fairness of comparison between transcriptional changes induced by the profiled compounds (which were all kinase inhibitors in this study) and the randomly sampled background lists. Every specimen was assigned a label of disease severity based on the following mapping to the Braak annotations[26]: A—early (Braak 0–2), B—intermediate (Braak 3–4), and C—late (Braak 5–6). The labels were used to establish three binary classification tasks, contrasting A vs. B, B vs. C, and A vs. C, with the expression of a prespecified list of genes playing the role of the input features.

Data for any given binary classification task were used to train prediction models using four different methods: logistic regression, support vector machines, boosted random forest models, and two-layer fully connected neural networks. The final ranking of drugs was derived from logistic regression models, because they exhibited the highest accuracy among the four machine learning methods (Supplementary Fig. 6). To address overfitting, a ridge regularization term that penalizes the L2-norm of feature weights was included in the models. No LASSO regularization was used, as it induces sparsity and excludes features that were specifically preselected to be included in the model.

Note that the proposed DRIAD framework is generally agnostic to the underlying method used to train classification models; importantly, the same method must be applied to random lists and the feature lists of interest. However, methods that guarantee that all preselected input features will be included in a model are more desirable, since they allow for an effective comparison of feature lists. As such, random forest models are not recommended, because one or more features may not get selected to be included in at least one decision tree.

Model performance was evaluated through leave-pair-out cross-validation. For a given binary classification task, each example in the dataset was associated with the example from the opposite class that was the closest match in age. If there were multiple candidates for the age match, the pairing was selected uniformly at random. The resulting set of age-matched pairs was evaluated in a standard cross-validation setting, by asking whether the later-stage example in each withheld pair was correctly assigned a higher score by the corresponding predictor. The fraction of correctly ranked pairs constitutes an estimate of the area under the ROC curve[85].

**Assessing gene list significance**. For a given gene list of interest, 1000 random gene lists of matching lengths were sampled from a uniform distribution over the protein-coding space. Additional analysis did not reveal any significant association of predictor performance to the pairwise correlations among selected genes, nor to the proximity of selected genes on a gene–gene interaction network. Based on these observations, we saw no reason to bias random gene selection toward more (or less) internal connectivity.

Gene lists produced by the random sampling constitute a background for comparison with a particular gene list of interest. After evaluating all lists through cross-validation, an empirical p-value was computed as the fraction of background lists that yield higher predictor performance than the gene list of interest. p-Values calculated for the same gene list across multiple datasets were combined to produce the HMP value[48]. Gene lists associated with post-perturbational transcriptional changes were sorted by HMP to produce the final ranking of the corresponding compounds.

**Assessing toxicity**. A multiwell cell dispenser (catalog# 5840300, Thermo Scientific,Waltham, MA) with standard tubing (catalog# 24072670, ThermoFisher Scientific, Waltham, MA) was used to plate 10,000 cells per well of a 96-well cell culture plate (catalog# 3603, Corning, Corning NY). Cells were treated with compounds (Supplementary Data 3) or DMSO using a D300 Digital Dispenser (Hewlett-Packard, Palo Alto, CA). D300 software was used to randomize dispensation of compounds. After 48 h, 60 µl of a solution containing 10% Optiprep (catalog# D1556, Millipore Sigma, St. Louis MO) diluted with PBS (catalog# 21-040-CV, Corning, Corning NY), and a 1:5000 dilution of Hoechst 33342 (catalog# H3570, ThermoFisher Scientific, Waltham, MA) was gently added to the side of each well using a multichannel pipette (catalog# 1060-0850, VistaLab Technologies, Brewster, NY) on the lowest speed setting of 1. After 30 min, 80 µl of a solution containing 3.7% formaldehyde (catalog# 15711, Electron Microsopy Sciences, Hatfield, PA), 20% Optiprep in PBS was added. After a 30-min incubation, a multichannel pipette (catalog# 1060-0850, VistaLab Technologies, Brewster, NY) was used to remove all but 15 µl from each well; 100 µl of 1× PBS was added to each well and the plate was covered with a foil seal (catalog# MSF-1001, Bio-Rad, Hercules, CA). Images were taken on an InCell 6000 (GE Healthcare Bio-Sciences, Pittsburgh, PA). Columbus Image Data Storage and Analysis System (Perkin Elmer, Waltham MA) was used to count the number of Hoechst stained nuclei as a readout of cell number.

**TAS, top targets, and polypharmacology analysis**
*Systematic classification of compound-target affinities using TAS*. Drug affinity data from ChEMBL v25 (ref. [86]) and in-house data comprising drug affinity curves, single-dose binding data from the DiscoverX platform and manual binding assertions curated from literature were compiled into a single consistent measure of binding affinity. Multiple measurements for the same drug–target combination were aggregated by calculating the first quartile. For each drug–target pair, we only considered the highest quality source of data. If full dose–response affinity

measurements were available, they took precedence over single-dose binding measurements, which took precedence over binding assertions mined from the literature.

Dose–response affinity data were converted to TAS scores based on empirically derived concentration cutoffs (<100 nM: TAS = 1; 100–999 nM: TAS = 2; 1–10 μM: TAS = 3, and >10 μM: TAS = 10). For single-dose drug binding data, we used concentration-specific thresholds derived from the empirical correlation between dissociation constant and percent inhibition (100 nM <25%: TAS = 2, ≥75%: TAS = 10; 1 μM <1%: TAS = 2, ≥90%: TAS = 10; 10 μM <0.1%: TAS = 2, ≥75%: TAS = 10). Drug–target pairs were assigned TAS = 2 or TAS = 10 if they were mentioned in confirming (e.g., drug X was equipotent for Y) or negative (e.g., drug X was found to not inhibit Y) statements in the literature.

All TAS values used in this study are based on v25 of the Small Molecule Suite, which is publicly available through https://smallmoleculesuite.org.

*Identification of important target genes using TAS profiles.* Drugs were ranked based on their HMP scores, as computed above. If a drug was profiled in more than one 3′-DGE experiment, the corresponding HMP scores were averaged with a geometric mean. For a given target of interest, ECDFs were computed for each TAS value separately, using the HMP-based ranking as input. Area under individual ECDFs provides a summary statistic for the overall placement in the ranking of drugs with the corresponding TAS value.

The importance of a particular drug target was assessed through the one-sided Kendall's Tau test, which compares whether pairs of drugs are ordered the same way in two different rankings. In our case, a pair of drugs is considered to be concordant if the drug with the higher binding affinity (lower TAS value) appears closer to the front of the HMP-based ranking, and discordant otherwise. The Kendall's Tau coefficient is then defined as the fraction of concordant pairs among those that can be ordered (i.e., pairs of drugs with non-identical TAS values). The associated p-value of the Tau coefficient is approximated through standard permutation testing. Targets with fewer than three confirmed binders (TAS = 1, 2, 3) were not evaluated.

*Polypharmacology analysis of target gene combinations.* We first compiled a list of pairwise gene combinations for which we had TAS scores for both targets from at least six compounds, to ensure that enough points were available for meaningful Wilcoxon rank sum tests. For each combination of targets, we split the compounds into three categories based on their TAS scores: compounds that bind both targets (category "A AND B") and compounds that bind one of the targets but not the other (categories "A AND NOT B" and "B AND NOT A"). A union of the latter two was defined to be an "A XOR B" set. Wilcoxon rank sum tests were performed separately for "A AND NOT B", "B AND NOT A", and "A XOR B". In all cases, the comparison was made relative to "A AND B".

For each target pair, p-values from individual Wilcoxon rank sum tests were averaged using the harmonic mean[48]. The resulting p-values were further aggregated using the Brown's method (an extension of the Fisher's method) with the test dependence metric being defined as the Jaccard similarity of the corresponding compound sets. For example, DCLK3 participated in 250 pair evaluations. Two of those evaluations included positive interactions "DCLK3 AND DYRK1B" and "DCLK3 AND DAPK3" pairings, with the corresponding HMP values 0.00099 and 0.0015, respectively (Fig. 5b). There are ten compounds binding DCLK3 and DYRK1B, and the same ten compounds also bind to DCLK3 and DAPK3, yielding a Jaccard similarity of 1.0 for the two pairings. The two p-values are thus considered to be coming from entirely non-independent tests by the Brown's method, which aggregates all 250 p-values into a single metric of importance for DCLK3 (Fig. 5c).

**Reporting summary**. Further information on research design is available in the Nature Research Reporting Summary linked to this article.

## Data availability

Raw post-perturbational gene expression data for the 80 compounds profiled in this study, the associated gene lists, drug toxicity data, and all relevant metadata have been uploaded to Synapse[87]. Additional datasets analyzed during the current study are available in the Synapse repository, https://adknowledgeportal.synapse.org/. Raw sequencing data are additionally available at the NCBI Sequence Read Archive (https://www.ncbi.nlm.nih.gov/sra/?term=SRP301436). Processed data are available on Gene Expression Omnibus (https://www.ncbi.nlm.nih.gov/geo/query/acc.cgi?acc=GSE164788).

## Code availability

The machine learning framework for evaluating the capacity of gene lists to predict disease severity is publicly available as an R package[88]. A web application that allows users to evaluate their own gene lists on the different datasets, brain regions, and binary classification tasks is available at https://labsyspharm.shinyapps.io/DRIAD/. Scripts to fully reproduce the tables and figures presented in this manuscript are provided on GitHub at https://github.com/labsyspharm/DRIADrc. The reproducibility was made possible in part by the R packages grImport2 (ref. [89]) and gridSVG[90].

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

## Acknowledgements

We acknowledge support from the NIA grant R01 AG058063 (S.R., C.H., P.T., A.S., M.W.A., B.H., and P.K.S.), U54-CA225088 and a supplement CA225088-02S1 (S.R., M.W.A., N.T.J., K.E., P.K.S.), U24-DK116204 (N.M., S.A.B., and P.K.S.) the CART fund (awarded to M.W.A.), and by Harvard Catalyst Program for Faculty Development and Diversity Inclusion (PFDD) Faculty Fellowship (awarded to S.R.). The results published here are in part based on data obtained from the AMP-AD Knowledge Portal (https://doi.org/10.7303/syn2580853). These data were generated from postmortem brain tissue collected through the Mount Sinai VA Medical Center Brain Bank, led by Dr. Eric Schadt from Mount Sinai School of Medicine, by the Rush Alzheimer's Disease Center, Rush University Medical Center, Chicago, and by the following sources: The Mayo Clinic Alzheimer's Disease Genetic Studies, led by Dr. Nilufer Taner and Dr. Steven G. Younkin, Mayo Clinic, Jacksonville, FL, using samples from the Mayo Clinic Study of Aging, the Mayo Clinic Alzheimer's Disease Research Center, and the Mayo Clinic Brain Bank. We would like to thank Sudeshna Das, Colin Magdamo, Roy Welsch, Stan Finkelstein, Ioanna Tzoulaki, Deborah Blacker, and Lefkos Middleton for their insightful comments and suggestions, and Paul Murrell for his relentless help with grImport2 and gridSVG packages that enabled complete figure reproducibility within R.

## Author contributions

S.R., S.B., K.E., and g.Z. collected the transcriptional perturbation and drug toxicity data. P.T. and A.S. implemented the machine learning framework. C.H. and N.M. conducted the drug target analysis. All authors contributed to figures and manuscript writing.

## Competing interests

The authors declare the following competing interests. P.K.S. is a member of the SAB or Board of Directors of Applied Biomath, RareCyte, NanoString and Glencoe Software and has equity in some of these companies. In the last 5 years, the Sorger lab has received research funding from Novartis and Merck. P.K.S. declares that none of these relationships are directly or indirectly related to the content of this manuscript. B.T.H. has stock in Novartis and Dewpoint. N.T.J. is an employee of H3 Biomedicine, a subsidiary of Eisai Inc. that develops therapies for Alzheimer's. S.R., P.K.S., M.W.A., and A.S. are inventors on a patent application (WO/2017/173451) for novel targets in neurodegenerative diseases. All other authors (C.H., P.T., N.M., S.B., K.E., G.Z.) declare no competing interests.
