## [Peer Review File · Nature Communications]

Reviewers' Comments:

Reviewer #1:

Remarks to the Author:

Steve Rodriguez et al., proposed to use a machine learning framework (DRIAD) to repurpose drugs for Alzheimer's disease. The key datasets included gene expression data in AMP-AD database using AD brain tissues with different stages of disease severity, as well as the experimental data from drugs induced gene expression perturbations. They found that top-scoring drugs were inspected for common trends among their nominal molecular targets, suggesting the Janus (JAK), Unc-51-like (ULK) and NIMA-related (NEK) families as potential targets. Although the methodology is original, the idea of linking AD related gene expression and drug induced gene expression is not very novel, please see these references (PMID: 29959820; PMID: 31492831; PMID: 28005991).

There is no doubt that this is an important area of establishing strategies of repurposing existing drugs for AD, but I do have several concerns regarding both the dataset used in the study, as well as the strategies used in the current manuscript.

Major comments:

1. In AD research, we are usually interested in the factors associated with disease risk, as well as factors linked to disease severity or age of onset (or phenotype modifiers). Here, the author did not specify why only gene-expression data associated with AD Braak stages were used. It is of course important to find those phenotype modifiers, but it is also important for those genes associated with AD disease risk. That means, case-control gene-expression data in AD brain tissues should also be included.

2. The authors used the AMP-AD database for gene expression dataset in AD brains. It is important to address why only AMP-AD database was included, instead of other raw data reported in several other publications. In addition, it will be important to describe the clinical information of each brain samples cohort involved in the current analysis. For example, which brain regions were selected for RNA-seq, what's the pathology of the tissues (eg. Tau, Amyloid beta), how many patients were analyzed, and what type of assays were used for measuring gene expressions?

3. Machine learning is a general concept, which may include neural network, deep learning etc. A classifier, can also include many different methods. While logistic regression is just one type of them. I think that using machine learning in the tile might not be very precise/appropriate.

4. The authors mentioned that "We found that most of the published lists of AD-associated genes outperformed randomly-selected lists of equal length (Fig. 1c). Thus, DRIAD effectively recapitulates previous attempts to identify genes and co-expression modules associated with disease severity."

I could not find the list of AD-associated genes in the current manuscript. SO what are these genes, and did authors test some well-known AD genes, such as APP, PSEN1, PSEN2, TREM2, APOE. and how do those genes work in the DRIAD framework.

5. The current study included only 80 drugs in the drugs induced gene expression study. Majority of them are kinase. SO why were this specific group of drugs included for targeting kinases? There are many other proteins/genes are important in AD pathological process, eg. Some cell surface proteins (such as TREM2, and CD33) .

6. It is impressive that the study assessed both toxicity and affinity of the drugs to targets (and specificity). I think another important question is that could those drugs cross blood-brain barrier (BBB)? It should also be considered in this drug repurposing framework.

7. In several paragraph of the manuscript, the authors mentioned "drug-induced perturbation of neurons". However, it is in fact not just neurons, it is a mixed culture of neurons, glia and oligodendrocytes in the 3DGE test; as described in methods.

8. The DRIAD used both AD stages-gene expression and drugs induced gene expression data. But it is unclear, when the top candidates were selected, does it also consider if the trend of gene expression caused by drugs and disease status are consistent for potential therapy? For example, if gene A is increased in a severe stage of AD, and drug B also increase gene A expression; using drug B may not treating AD, but make it worse.

Minor comments:

1. Some typos in the text

"Discussion: In this paper we described the development of DRIAD, a machine learning framework for evaluating potential relationships between a disease and any biological process than can be described by a list of genes."

"Than" should be that

2. There is no titles for Supplemental table. It is difficult for readers to understand what does each column mean. what was the adjustment for.

3. Page-5 "To validate the DRIAD framework, lists of gene names previously reported in the literature to be associated with an aspect of AD progression^{24,26-33} were substitute for DGLs."

What are the lists of genes included in the training?

"We found that most of the published lists of AD-associated genes outperformed randomly-selected lists of equal length (Fig. 1c)."

What percentage of genes outperformed the randomly selected lists?

"Similarly, Mostafavi, et al., identified a co-expression module of 390 genes that has a strong association with cognitive decline"

Why selecting these 390 genes for training? How to validate that there is enough power to establish the classifier.

Reviewer #3:

Remarks to the Author:

Drug repurposing is an interesting opportunity, and the paper presents a novel and innovative approach which looks exciting. I am therefore very supportive of the paper with a few caveats

1/There is other literature regarding transcriptional profiles as a means of identifying compounds for repositioning as AD therapies eg Williams et al Drug repurposing for Alzheimer's disease based on transcriptional profiling of human iPSC-derived cortical neurons Transl Psychiatry. 2019; 9:

220. Published online 2019 Sep 6. doi: 10.1038/s41398-019-0555-x It would be good to discuss other transcriptional approaches.

2/ The limitations of not separately validating the compounds in AD in vitro models to examine impacts on key pathologies etc should be acknowledged

3/The paper could perhaps be shortened and more punchy in places

Reviewer #4:

Remarks to the Author:

Review on the manuscript entitled "Machine Learning Identifies Novel Candidates for Drug Repurposing in Alzheimer's Disease", by Rodriguez et al.

The authors are presenting a computational framework that quantifies potential associations between the pathology of AD severity and molecular mechanisms. They proceeded with a validation of their method and then they applied it to evaluate 80 drugs and compounds as possible repurposed drug candidates.

Comments:

The Introduction is well-structured. However, I find the paragraph regarding drug repurposing tools/databases incomplete. I would suggest the authors to refer to more recent and of wide use tools. I would suggest to take ideas from papers like: Sam E, Athri P. Web-based drug repurposing tools: a survey. Brief Bioinform. 2019;20(1):299-316. doi:10.1093/bib/bbx125

and to refer to tools from the following links:

<http://www.lincsproject.org/LINCS/tools>

<http://lincs-dcic.org/>

<https://amp.pharm.mssm.edu/L1000CDS2/#/index>

<https://amp.pharm.mssm.edu/l1000fwd/>

<http://amp.pharm.mssm.edu/CREEDS/>

<https://clue.io/repurposing>

The authors mention that a drawback of the existing repurposing databases is that they are obtained from diverse biological settings. This is true, however, an even bigger problem is that this diversity is not balanced. There is a clear bias on data coming from cancer cell lines and biopsies that should be mentioned.

Also I am finding a contradiction: the authors say that another drawback of the existing repurposing tools is that they are rarely disease specific. However, at the end of the Introduction they are saying that DRIAD is agnostic to the type of the disease.

In the Methods:

* When profiling using 3'DGE, the authors keep the top 300 differentially expressed genes each time ensuring that <2% of the transcription space is captured. This is not a strong argument for this threshold, since one could select 1% (150 genes) under the same argument. Also, is it necessary to have constant list length? What is the fold change range for the selected genes? What is the proportion between the over and under expressed genes each time?

* I find that there is a bias in the drug induced gene expression perturbation since the drugs belong

only to kinase inhibitors class. The authors should elaborate more on this.

*The selection of logistic regression classifier "due to its popular use with gene expression data" is not a sufficient reason. The same counts for the need of having preselected input features. Since the selection of the list of 300 genes is quite arbitrary, this arbitrariness is propagated by

keeping

a constant length for the feature vector. It would be more reasonable to have a feature (gene) selection from each list, thus creating sublists with bigger significance. The comment on random forests is too specific since there are many more (and different)

learning methods that could work fine (or even better) in this pipeline.

I think that by using a logistic regression classifier with preselected input feature vector, given that the preselection method is not robust and well-explained, the authors miss the real power of Machine Learning (ML).

And this makes the title of the paper quite misleading.

Under the same note, the authors say that "the choice of a classification method does not matter" if a comparison

with randomly selected features is done. This is a quite unsupported statement. Perhaps this is valid

when comparing between random lists and the lists of interest. However, in the regular classification run and not

in the validation stage, the selection of the ML methodology is of major importance and characterizes the pipeline.

* The performance evaluation of the model using leave-pair-out cross validation is a weak type of evaluation.

How many samples were used? It should be clearly stated. The recommended way is to start with a leave-one-out cross validation

and then proceed with evaluation on an independent set.

* In the TAS section, I would suggest the authors to discuss the possible effect in the results that the different quality of binding data has, since there is quite a lot heterogeneity in the retrieval of binding data

for each drug-target combination.

* In the polypharmacology section what is the rationale behind the selection of six compounds?

* What are the selection criteria for the 80 compounds? Are sufficiently chemically diverse? Did the authors

try to cluster them based on their chemical structure?

* In Table 2 Caption: Please refer to the headers in the order they are shown in the table. Also please refer to p_{adj} and display

the p_{adj} in a uniform number format.

* In Figure 5: How is the density estimate calculated? What are the axes units and range in figure 5 (b) ?

In the Results:

* The success of the classifier is based on the comparison of the input features with regards to the randomly selected features.

This is not a strong evidence since the demonstrated AUCs are not remarkably high (<0.85) as shown in figures 1 and 2.

* The authors state that a drug's "mechanism" is encapsulated in the DGL. This is a weak hypothesis since in the DGL there is a multiplex of mechanisms along with a number of genes that are included in the DGL by chance (gene expression noise).

* The size of the list is expected to play a role in the classification. I would suggest to perform classification with incrementally increasing size till the finally selected size.

* I find very interesting the TAS analysis and the Polypharmacology Analysis. However, the final results

regarding candidate drugs and mechanisms related to AD are quite generic and of limited interest. There is not enough evidence

for further investigation on them.

* In lines 296-298, the 80 compounds are analyzed in 35 FDA-approved, 35 pre-clinical and 15

investigational.

However this sums up to 85. Please correct.

REVIEWER COMMENTS

Reviewer #1 (Remarks to the Author):

Steve Rodriguez et al., proposed to use a machine learning framework (DRIAD) to repurpose drugs for Alzheimer's disease. The key datasets included gene expression data in AMP-AD database using AD brain tissues with different stages of disease severity, as well as the experimental data from drugs induced gene expression perturbations. They found that top-scoring drugs were inspected for common trends among their nominal molecular targets, suggesting the Janus (JAK), Unc-51-like (ULK) and NIMA-related (NEK) families as potential targets. Although the methodology is original, the idea of linking AD related gene expression and drug induced gene expression is not very novel, please see these references (PMID: 29959820; PMID: 31492831; PMID: 28005991).

Response R1-0: We thank the reviewer for pointing out these relevant studies (one of which we did cite originally). We have expanded our overview of previous drug repurposing approaches to incorporate the additional references suggested by the three reviewers.

There is no doubt that this is an important area of establishing strategies of repurposing existing drugs for AD, but I do have several concerns regarding both the dataset used in the study, as well as the strategies used in the current manuscript.

Major comments:

- 1. In AD research, we are usually interested in the factors associated with disease risk, as well as factors linked to disease severity or age of onset (or phenotype modifiers). Here, the author did not specify why only gene-expression data associated with AD Braak stages were used. It is of course important to find those phenotype modifiers, but it is also important for those genes associated with AD disease risk. That means, case-control gene-expression data in AD brain tissues should also be included.*

Response R1-1: We agree with the reviewer's perspective that this approach could be applied to the risk of acquiring AD through the analysis of case control studies. We are in part addressing this already, because patients with Braak stage I (and in some cases Braak stage II) are unlikely to demonstrate clinical symptoms. From a clinical standpoint, these individuals would be considered a control. From a pathologic perspective, gene expression data sets from age-matched individuals with no evidence of AD pathology could be added as "controls" to build a predictor for the risk of AD pathology. However, this would constitute a separate study, and we believe that it is outside the scope of the current work. We have added this point in the discussion as part of future directions.

- 2. The authors used the AMP-AD database for gene expression dataset in AD brains. It is important to address why only AMP-AD database was included, instead of other raw data reported in several other publications. In addition, it will be important to describe the clinical information of each brain samples cohort involved in the current analysis. For example, which brain regions were selected for RNA-seq, what's the pathology of the tissues (eg. Tau, Amyloid beta), how many patients were analyzed, and what type of assays were used for measuring gene expressions?*

Response R1-2: The AMP-AD Knowledge Portal is the most comprehensive database of gene expression profiles from AD brains, combining data from multiple large-scale studies. Given the already lengthy description of our analyses and results, we chose to exclude other smaller-scale studies. However, it is important to recognize that the methodology is general enough to be applied to other data sources and data types (e.g., mass spec proteomics, genome-wide association studies, etc.). We included a brief note of this in Discussion.

With respect to the point about specifying brain regions, this is shown in Figure 1b. Per reviewer's request, we included additional annotations showing the number of samples captured in each data slice. We also clarified in the text that Braak staging was assigned through neuropathological assessment of neurofibrillary tangle accumulation.

- 3. Machine learning is a general concept, which may include neural network, deep learning etc. A classifier, can also include many different methods. While logistic regression is just one type of them. I think that using machine learning in the title might not be very precise/appropriate.*

Response R1-3: We thank the reviewer for bringing up a very important point. In our preliminary analysis, we had assessed several different machine learning methods and settled on logistic regression because we had observed that it yielded higher performance on average. We now formally include additional machine learning methods in the DRIAD framework (<https://github.com/labsyspharm/DRIAD>) with a README explaining how the user may choose which method to run. We also include a new **Supplementary Figure 6**, showing how model performance varies across the different methods. Given this expansion in scope we believe that is appropriate to leave "machine learning" in the title.

- 4. The authors mentioned that "We found that most of the published lists of AD-associated genes outperformed randomly-selected lists of equal length (Fig. 1c). Thus, DRIAD effectively recapitulates previous attempts to identify genes and co-expression modules associated with disease severity." I could not find the list of AD-associated genes in the current manuscript. SO what are these genes, and did authors test some well-known AD genes, such as APP, PSEN1, PSEN2, TREM2, APOE. and how do those genes work in the DRIAD framework.*

Response R1-4: We apologize for not making it clear that the gene sets collected from the literature were released along with all other data on www.synapse.org/DRIAD. To ensure that the manuscript is self-contained, we further wrangled these sets into a standalone Supplementary Table 2, which is now included in the supplement in .xlsx format. Furthermore, we deployed a web application (<https://labsyspharm.shinyapps.io/DRIAD/>) allowing users to run DRIAD on their favorite list of genes. For example, querying the web application with a gene set consisting of APP, PSEN1, PSEN2, TREM2, APOE shows that it significantly ($p=0.015$) outperforms randomly-selected five-gene sets when training a predictor to recognize intermediate (B) vs. late (C) disease stages in Mount Sinai Brain Bank gene expression data.

- 5. The current study included only 80 drugs in the drugs induced gene expression study. Majority of them are kinase. SO why were this specific group of drugs included for targeting kinases? There are many other proteins/genes are important in AD pathological process, eg. Some cell surface proteins (such as TREM2, and CD33).*

Response R1-5: The reviewer is correct of course. This is a proof-of-principle study that we expect to scale to all FDA-approved and bioactive collections. Our initial focus is on well-characterized compounds that are known to have relatively large effects – this particular set of kinase inhibitors was chosen to span kinase space in a compact collection. We agree with the reviewer that there are many other important genes and proteins relevant to the AD pathology, and our future experiments will focus on some them. In the case of TREM2, microglia cells are needed to measure the effect of its perturbation [PMC7136959, PMC5839761], which is not currently included in our mixed cell culture of differentiated neuroprogenitors. The only available therapeutic agent against CD33 is a cytotoxic antibody drug conjugate (Gemtuzumab) that induces efficient cell killing in some cell lines; we consider this to be an undesirable mechanism of action for a drug for neurodegeneration. In contrast, we used kinase inhibitors at doses that were not acutely cytotoxic.

- 6. It is impressive that the study assessed both toxicity and affinity of the drugs to targets (and specificity). I think another important question is that could those drugs cross blood-brain barrier (BBB)? It should also be considered in this drug repurposing framework.*

Response R1-6: We thank the reviewer for recognizing our efforts in quantifying the toxicity and target affinity of the drugs in our study. We strongly agree on the importance of knowing if drugs cross the blood-brain barrier (BBB) for their use in brain diseases. Unfortunately, unlike the data on toxicity and affinity, there are no databases containing information on the penetration of compounds across the BBB. Because these compounds were not developed to treat diseases of the brain identifying information on BBB penetration has been complicated and we find inconsistencies in the methodology and animal models used across studies. This is a source of frustration in the case of our most promising compounds. Theoretical approaches have been developed that aim to predict the penetration of drugs into the brain based on physical characteristics and chemical structure, however, the predictability is not generalizable across all compounds [PMID: 29556336]. Therefore, the only way to know if a drug crosses the BBB is through empirical studies, first in animals and then in patients, with the disease of interest. We envision that detecting drugs in cerebral spinal fluid (CSF), which can be collected from Alzheimer's patients by lumbar puncture, will be the most effective approach.

7. *In several paragraphs of the manuscript, the authors mentioned "drug-induced perturbation of neurons". However, it is in fact not just neurons, it is a mixed culture of neurons, glia and oligodendrocytes in the 3DGE test; as described in methods.*

Response R1-7: We thank the reviewer for pointing out the discrepancy – we were trying to be brief rather than inaccurate. All mentions of the cell culture used in our experiments have been corrected to its precise definition.

8. *The DRIAD used both AD stages-gene expression and drugs induced gene expression data. But it is unclear, when the top candidates were selected, does it also consider if the trend of gene expression caused by drugs and disease status are consistent for potential therapy? For example, if gene A is increased in a severe stage of AD, and drug B also increase gene A expression; using drug B may not treating AD, but make it worse.*

Response R1-8: Unfortunately, the directionality of the effect is not straightforward to quantify: every gene list contains both correlated and anti-correlated genes. Rather than counting how many genes have their expression increase or decrease by a given compound, we instead performed a cytotoxicity assay to determine if the compound is neurotoxic in a secondary screen (Fig 3). Additional safety and efficacy assessment will of course need to come from animals and then clinical trials.

Minor comments:

9. *Some typos in the text "Discussion: In this paper we described the development of DRIAD, a machine learning framework for evaluating potential relationships between a disease and any biological process than can be described by a list of genes. "Than" should be that*

Response R1-9: We thank the reviewer for catching this. The typo has been corrected.

10. *There is no titles for Supplemental table. It is difficult for readers to understand what does each column mean. what was the adjustment for.*

Response R1-10: We apologize for not being more explicit; the captions for supplementary tables were included in the main manuscript, which made them disconnected from the tables they were describing. We copied the captions to be directly in the corresponding .xlsx files.

11. *Page-5 "To validate the DRIAD framework, lists of gene names previously reported in the literature to be associated with an aspect of AD progression^{24,26–33} were substitute for DGLs."What are the lists of genes included in the training?*

Response R1-11: As mentioned in **Response R1-4**, we added Supplementary Table 2 which explicitly shows the gene lists collected from previously published studies of AMP-AD datasets.

12. *"We found that most of the published lists of AD-associated genes outperformed randomly-selected lists of equal length (Fig. 1c)." What percentage of genes outperformed the randomly selected lists?*

Response R1-12: We apologize for not being clear, but all genes in a given list should be treated as a single unit. We say that one list outperforms another if the corresponding predictor accuracy is higher. Every published gene list was compared against 100 randomly-selected lists, and the fraction of randomly-selected lists that outperform a given published list is given by the corresponding empirical p value presented in Fig 1c.

13. *"Similarly, Mostafavi, et al., identified a co-expression module of 390 genes that has a strong association with cognitive decline" Why selecting these 390 genes for training? How to validate that there is enough power to establish the classifier.*

Response R1-13: We apologize for not being clear in the manuscript. The 390 genes in question were identified by Mostafavi, et al., as being predictive of cognitive decline based on their analysis of AMP-AD datasets. Given that we are also working with AMP-AD data, and our study is focused on predicting the Braak stage (which is hypothesized to be correlated with cognitive decline), the 390-gene module defined by Mostafavi, et al., is a relevant positive control. The statistical power is defined by the 100 randomly-selected gene lists against which the 390-gene module is evaluated. We believe that a sample of 100 sets provide sufficient statistical power to derive the empirical p values shown in Fig. 1c. However, we would be happy to increase the number of randomly-selected gene sets if the reviewer deems it necessary.

Reviewer #3 (Remarks to the Author):

Drug repurposing is an interesting opportunity, and the paper presents a novel and innovative approach which looks exciting. I am therefore very supportive of the paper with a few caveats

1. *There is other literature regarding transcriptional profiles as a means of identifying compounds for repositioning as AD therapies eg Williams et al Drug repurposing for Alzheimer's disease based on transcriptional profiling of human iPSC-derived cortical neurons Transl Psychiatry. 2019; 9: 220. Published online 2019 Sep 6. doi: 10.1038/s41398-019-0555-x It would be good to discuss other transcriptional approaches.*

Response R3-1: This particular reference has also been suggested by Reviewer 1. As mentioned in **Response R1-0**, we expanded our overview of previous approaches to include additional studies.

2. *The limitations of not separately validating the compounds in AD in vitro models to examine impacts on key pathologies etc should be acknowledged.*

Response R3-2:

We thank the reviewer for bringing this limitation to our current study, which can be explored in subsequent studies. We add to the last paragraph of the discussion, "Our results require validation in relevant in vitro and in vivo AD model systems with amyloid plaques, neurofibrillary tangles, and neuronal death (PMID 29950669) to examine impacts on key pathologies."

3. *The paper could perhaps be shortened and more punchy in places*

Response R3-3: We thank the reviewer for their suggestion. We performed a full review of the manuscript, but were unable to identify sections that could obviously be shortened or removed without sacrificing clarity. However, we did edit extensively and tried to reduce word count while also addressing all changes requested by the referees. We are happy to consider specific suggestions.

Reviewer #4 (Remarks to the Author):

Review on the manuscript entitled "Machine Learning Identifies Novel Candidates for Drug Repurposing in Alzheimer's Disease", by Rodriguez et al.

The authors are presenting a computational framework that quantifies potential associations between the pathology of AD severity and molecular mechanisms. They proceeded with a validation of their method and then they applied it to evaluate 80 drugs and compounds as possible repurposed drug candidates.

Comments:

- 1. The Introduction is well-structured. However, I find the paragraph regarding drug repurposing tools/databases incomplete. I would suggest the authors to refer to more recent and of wide use tools. I would suggest to take ideas from papers like: Sam E, Athri P. Web-based drug repurposing tools: a survey. Brief Bioinform. 2019;20(1):299-316. doi:10.1093/bib/bbx125 and to refer to tools from the following links:
<http://www.lincsproject.org/LINCS/tools>
<http://lincs-dcic.org/>
<https://amp.pharm.mssm.edu/L1000CDS2/#/index>
<https://amp.pharm.mssm.edu/1000fwd/>
<http://amp.pharm.mssm.edu/CREEDS/>
<https://clue.io/repurposing>*

Response R4-1: We thank the reviewer for pointing out the survey of web-based tools. As mentioned in **Response R1-0**, we expanded our overview of previous drug repurposing approaches by including seven additional references suggested by one or more reviewers.

- 2. The authors mention that a drawback of the existing repurposing databases is that they are obtained from diverse biological settings. This is true, however, an even bigger problem is that this diversity is not balanced. There is a clear bias on data coming from cancer cell lines and biopsies that should be mentioned.*

Response R4-2: We agree with the reviewer and added a brief mention of the bias to the Introduction.

- 3. Also I am finding a contradiction: the authors say that another drawback of the existing repurposing tools is that they are rarely disease specific. However, at the end of the Introduction they are saying that DRIAD is agnostic to the type of the disease.*

Response R4-3: We apologize for the confusion; our intent was to communicate that the data considered and the models generated by DRIAD are disease-specific. However, the proposed methodology of gene list evaluation is general enough that similar frameworks can be established for other diseases. To improve clarity, we removed the statement that DRIAD is disease-agnostic from the Introduction.

In the Methods:

- 4. When profiling using 3'DGE, the authors keep the top 300 differentially expressed genes each time ensuring that <2% of the transcription space is captured. This is not a strong argument for this threshold, since one could select 1% (150 genes) under the same argument. Also, is it necessary to have constant list length? What is the fold change range for the selected genes? What is the proportion between the over and under expressed genes each time?*

Rebuttal Figure 4-4: The distribution of how many differentially expressed genes satisfied the FDR < 0.05 threshold across all compounds profiled by 3' Digital Gene Expression.

Response R4-4: We apologize for not making this clear in the manuscript, but drug-associated gene lists were composed from genes that were differentially expressed under an FDR < 0.05 cutoff. For most compounds, this produced a list with 300 **or fewer** genes (**Rebuttal Figure 4-4**). A small handful of compounds produced more substantial transcriptional changes, resulting in significant (FDR < 0.05) perturbation of as many as 3,000 genes. We chose to cap these lists at the top 300 genes (the natural “cliff” in the distribution) because this **a**) made them more consistent with the vast majority of drugs in the panel, increasing the fairness of comparisons across compounds, **b**) increased the sampling space for the corresponding background sets, and **c**) helped prevent overfitting that arises when the number of features is vastly larger than the number of samples, which in our case was on the order of several hundred (**updated Fig 1b**). We have clarified and expanded the corresponding description in the Methods section.

5. *I find that there is a bias in the drug induced gene expression perturbation since the drugs belong only to kinase inhibitors class. The authors should elaborate more on this.*

Response R4-5: As discussed in **Response R1-5**, this is an excellent point, and we do not mean to imply that this is the best or only class of compounds in which to find potential repurposing opportunities. Ours is a proof-of-principle study that we expect to scale to all FDA-approved and bioactive collections. The primary focus was on kinase inhibitors because they **a**) produce relatively large transcriptional changes, which can be effectively measured with high-throughput intermediate read density assays like the 3' DGE; **b**) have been extensively profiled, resulting in rich target information; and **c**) have been previously characterized in the literature, increasing the connection of our results to known biological and clinical contexts.

6. *The selection of logistic regression classifier "due to its popular use with gene expression data" is not a sufficient reason. The same counts for the need of having preselected input features. Since the selection of the list of 300 genes is quite arbitrary, this arbitrariness is propagated by keeping a constant length for the feature vector. It would be more reasonable to have a feature (gene) selections from each list, thus creating sublists with bigger significance. The comment on random forests is too specific since there are many more (and different) learning methods that could work fine (or even better) in this pipeline. I think that by using a logistic regression classifier with preselected input feature vector, given that the preselection method is not robust and well-explained, the authors miss the real power of Machine Learning (ML). And this makes the title of the paper quite misleading. Under the same note, the authors say that "the choice of a classification method does not matter" if a comparison with randomly selected features is done. This is a quite unsupported statement. Perhaps this is valid when comparing between random lists*

and the lists of interest. However, in the regular classification run and not in the validation stage, the selection of the ML methodology is of major importance and characterizes the pipeline.

Response R4-6: As discussed in **Response R4-4**, the feature vectors are not constant in length. Instead, they comprise significantly differentially expressed genes, as being suggested by reviewer here. We agree with the reviewer that other machine learning methods should be part of DRIAD framework (and we in fact explored many of them in preliminary studies). We have therefore added support for random forests, neural networks and support vector machines in the DRIAD implementation (<https://github.com/labsyspharm/DRIAD>) with a README explaining how to choose which method to run. We have also included a new **Supplementary Figure 6**, showing how the different ML methods compare on gene lists associated with the compounds we profiled. Lastly, we revised the corresponding Methods subsection to better explain how method selection may affect the comparison of gene lists.

- 7. The performance evaluation of the model using leave-pair-out cross validation is a weak type of evaluation. How many samples were used? It should be clearly stated. The recommended way is to start with a leave-one-out cross validation and then proceed with evaluation on an independent set.*

Response R4-7: We completely agree that the standard way to evaluate predictors is with a nested cross-validation scheme, where the inner validation loop is used for parameter tuning, while the outer loop performs evaluation on a previously unseen data slice that was not used in model selection and training. However, our training included no parameter tuning, which removed the need for the inner cross-validation loop. We chose leave-pair-out cross-validation (LPOCV) for the main evaluation loop because **a)** LPOCV has less bias in its estimate of performance than all other popular cross-validation schemes, including leave-one-out cross-validation [<https://doi.org/10.1016/j.csda.2010.11.018>], and **b)** LPOCV allows for pairs of test samples to be age-matched to ensure that predictors are evaluated on their ability to predict the disease stage and not age, which is its confounder.

- 8. In the TAS section, I would suggest the authors to discuss the possible effect in the results that the different quality of binding data has, since there is quite a lot heterogeneity in the retrieval of binding data for each dru-target combination.*

Response R4-8: We agree that integrating binding data derived from multiple types of measurements and sources of different quality results in some uncertainty about the precision of each individual measurement. However, we believe that the more comprehensive target coverage provided by TAS scores outweighs the disadvantage of lower precision for our purposes. We aim to identify targets that are consistently enriched in high-scoring compounds, therefore the precision of each individual measurement is less important than in other applications. TAS vectors also attempt to correct for various types of research biases. We have added a brief note about this tradeoff in the manuscript.

- 9. In the polypharmacology section what is the rationale behind the selection of six compounds?*

Response R4-9: We apologize for not being clear, but the filter is “*at least* six compounds”. The lower bound is needed to ensure that a sufficient number of points is available for meaningful Wilcoxon rank sum tests. We clarified this point in the corresponding Methods subsection.

- 10. What are the selection criteria for the 80 compounds? Are sufficiently chemically diverse? Did the authors try to cluster them based on their chemical structure?*

Response R4-10: The compounds were selected such that they had readily available target information (<https://smallmoleculesuite.org>) and/or relevant published studies -- particularly in the case of FDA-approved compounds -- that would have increased the interpretability of our results. As suggested by the reviewer, we clustered our compounds based on their chemical structure with results included as a new **Supplementary Figure 5**. The vast majority of pairwise similarities were below 0.2, which was shown in our previous study to correlate with sufficient chemical diversity to efficiently span the kinome with a small kinase library [PMID: 30956147].

11. *In Table 2 Caption: Please refer to the headers in the order they are shown in the table. Also please refer to padj and display the padj in a uniform number format.*

Response R4-11: We thank the reviewer for their suggestion. The requested changes to Table 2 have been implemented.

12. *In Figure 5: How is the density estimate calculated? What are the axes units and range in figure 5 (b) ?*

Response R4-12: The density is calculated using the standard approach of convolving the data with a unit Gaussian kernel. The y-axis (Density) values are not as important as where the distribution peaks occur along the x axis (Drug Rank). We therefore normalized the density values of each distribution to allow for a better visual comparison of peak locations. The x axis is already annotated as "Drug Rank". We further annotated the y axis with "Normalized density estimate".

In the Results:

13. *The success of the classifier is based on the comparison of the input features with regards to the randomly selected features. This is not a strong evidence since the demonstrated AUCs are not remarkably high (<0.85) as shown in figures 1 and 2.*

Response R4-13: We agree with the reviewer that classifier accuracy is lower than what is generally considered to be "successful". However, low accuracy does not preclude the evaluation and comparison of feature sets (unless the classifiers are effectively random, which is not the case here). If a classifier trained on the expression of genes associated with a particular drug is substantially more accurate than equivalent classifiers trained on the expression of any arbitrarily chosen genes, then such a result suggests that the drug-associated genes carry at least some disease-related signal. Thus, instead of focusing on the raw accuracy values, DRIAD systematically quantifies how those accuracy values change with respect to the underlying feature sets.

14. *The authors state that a drug's "mechanism" is encapsulated in the DGL. This is a weak hypothesis since in the DGL there is a multiplex of mechanisms along with a number of genes that are included in the DGL by chance (gene expression noise).*

Response R4-14:

We agree with the reviewer that each drug acts by a multiplex of mechanisms. Mechanisms of action may also differ between cell types. This is particularly relevant in human neural cells given the heterogeneity of cell types in the human brain. The DGL reduces this complexity of cell types and mechanisms to a list of genes that can be compared to the heterogeneity and complexity of AD pathology. The ratio between the drug-related signal and gene expression noise is effectively assessed by comparing the DGL against equivalent lists composed entirely of noise in the form of randomly-selected genes. DGLs that substantially outperform randomly-selected lists are then treated as having high mechanism signal.

15. *The size of the list is expected to play a role in the classification. I would suggest to perform classification with incrementally increasing size till the finally selected size.*

Response R4-15: The reviewer brings up a very important point that classifier accuracy depends on the size of the feature set used to train it. In Supplementary Fig. 1, we show how predictor performance changes with incrementally increasing size of the gene list. In particular, we observe that prediction accuracy tends to increase, as the size of gene sets grows. To ensure fairness of comparisons, DGLs were always matched against randomly-selected gene lists of equal size.

16. *I find very interesting the TAS analysis and the polypharmacology Analysis. However, the final results regarding candidate drugs and mechanisms related to AD are quite generic and of limited interest. There is not enough evidence for further investigation on them.*

Response R4-16: We respectfully disagree with the reviewer. Previous studies [PMID: 25391662, PMID: 30792577], including our own, show that cytoplasmic double-stranded RNA (dsRNA) does accumulate in AD brains. In those studies, the presence of dsRNA has been linked to increased Type I interferon signaling, and it is encouraging to see several JAK inhibitors appear near the top of our ranked list. Taken together, the findings suggest that molecules suppressing Type I interferon signaling may carry some therapeutic potential, and we are further investigating this hypothesis in animal models, in observational electronic health record studies, and in a proof-of-concept clinical trial.

17. In lines 296-298, the 80 compounds are analyzed in 35 FDA-approved, 35 pre-clinical and 15 investigational. However this sums up to 85. Please correct.

Response R4-17: Five compounds were profiled twice (Supplementary Fig. 2), and we inadvertently double-counted them when reporting the counts on lines 296-298. We thank the reviewer for catching the discrepancy, which has now been corrected.

Reviewers' Comments:

Reviewer #1:

Remarks to the Author:

I'm satisfied by the authors' work to reply my comments. I'd still recommend to extend more discussion for some points (eg. BBB, other types of drugs targeting key AD genes, limitations of the current study/method). It would inform both basic scientists and clinical scientists about what will be next directions, and how could we work/improve in the field to narrow down the gap between computational prediction and drug discovery/clinical needs.

Reviewer #4:

Remarks to the Author:

The authors have successfully addressed my comments and suggestions. I have no further comments on the manuscript.

We appreciate the additional review conducted by the referees and their constructive feedback. Our point-by-point response can be found below.

Reviewer #1 (Remarks to the Author):

I'm satisfied by the authors' work to reply my comments. I'd still recommend to extend more discussion for some points (eg. BBB, other types of drugs targeting key AD genes, limitations of the current study/method). It would inform both basic scientists and clinical scientists about what will be next directions, and how could we work/improve in the field to narrow down the gap between computational prediction and drug discovery/clinical needs.

We extended the last paragraph of Discussion to focus on additional points that include the limitations of the current study (e.g., the lack of BBB penetration information), and the next steps needed to validate our computational “hits” experimentally and clinically.

Reviewer #4 (Remarks to the Author):

The authors have successfully addressed my comments and suggestions. I have no further comments on the manuscript.

We would like to express our gratitude to the reviewer for their extensive comments in the previous round of review. Their suggestions have substantially improved the manuscript.